# Reading signatures of supermassive binary black holes in pulsar timing array observations

Boris Goncharov [1,2] ✉, Shubhit Sardana [1,3], Alberto Sesana [4,5,6], Sharon Mary Tomson [1,2], John Antoniadis[7,8], Aurelien Chalumeau [9], David J. Champion [8], Siyuan Chen [10,11], Evan F. Keane [12], Kuo Liu[10,11], Golam Shaifullah [4,5,13], Lorenzo Speri [14] & Serena Valtolina [1,2]

Constraining the origin of the nanohertz gravitational-wave background necessitates precise noise modelling to avoid parameter estimation biases. In this work, we find the inferred properties of the putative gravitational wave background in the second data release of the European Pulsar Timing Array to be in better agreement with theoretical expectations under the improved noise model. In particular, our improved noise models show consistency of the background's strain spectral index with the value of −2/3, favoring the population of supermassive black hole binaries as the origin of the background. Our results further suggest that the observed gravitational wave emission is the dominant source of the binary energy loss, with no evidence of environmental effects or eccentric orbits. At the reference gravitational wave frequency of yr$^{-1}$, we also find a lower power-law strain amplitude of the background than in previous data analyses. This mitigates some of the tensions of the strain amplitude with the expected number density and mass scale of binaries discussed in the literature. Our analysis demonstrates the importance of accurate modelling of radio pulsar pulse profile variations, hierarchical properties of noise across pulsars, as well as noise model averaging, when inferring properties of the gravitational wave background.

Since 2020, Pulsar Timing Arrays (PTAs) have reported growing evidence for the nanohertz-frequency gravitational wave background in their data. The first tentative evidence came from a temporally correlated stochastic process in pulsar timing data[1–4]. The Fourier spectrum of delays and advances in pulsar pulse arrival times exhibited the expected spectral properties of the background. Most recently, PTAs —

with varying levels of statistical significance[5–8] — showed that this stochastic process exhibits Hellings-Downs correlations[9] consistent with the isotropic unpolarised stochastic gravitational wave background.

Properties of the gravitational wave background are assessed using the power law model of its characteristic strain spectrum:

[1]Max Planck Institute for Gravitational Physics (Albert Einstein Institute), Hannover, Germany. [2]Leibniz Universität Hannover, Hannover, Germany. [3]Department of Physics, IISER Bhopal, Bhopal, India. [4]Dipartimento di Fisica "G. Occhialini", Universitá degli Studi di Milano-Bicocca, Milano, Italy. [5]INFN, Sezione di Milano-Bicocca, Milano, Italy. [6]INAF - Osservatorio Astronomico di Brera, Milano, Italy. [7]FORTH Institute of Astrophysics, N. Plastira 100, Heraklion, Greece. [8]Max-Planck-Institut für Radioastronomie, Bonn, Germany. [9]ASTRON, Netherlands Institute for Radio Astronomy, PD Dwingeloo, The Netherlands. [10]Shanghai Astronomical Observatory, Chinese Academy of Sciences, Shanghai, China. [11]State Key Laboratory of Radio Astronomy and Technology, Chaoyang District, Beijing, P. R. China. [12]School of Physics, Trinity College Dublin, College Green, Dublin 2, Ireland. [13]INAF - Osservatorio Astronomico di Cagliari, Selargius, CA, Italy. [14]Max Planck Institute for Gravitational Physics (Albert Einstein Institute), Potsdam, Germany. ✉e-mail: boris.goncharov@me.com

$h_c(f) = A(f/\text{yr}^{-1})^{-\alpha}$. Here, $A$ is the strain amplitude at the gravitational wave frequency $f$ of $f_{\text{ref}} = \text{yr}^{-1}$, and $-\alpha$ is the power law spectral index. The corresponding spectral index of the power spectral density [$s^3$] of delays(-advances) [s] induced by the background in the timing data is $-\gamma$. These stochastic timing delays resulting from the gravitational-wave redshift of pulsar spin frequency are referred to as temporal correlations.

Supermassive black hole binaries at subparsec separations are expected to be a dominant source of the stochastic gravitational wave background at nanohertz frequencies[10]. However, the expected amplitude of the background is lower than the observations suggest[11]. This is visible in Figure 7 from ref. 12 and Fig. 5 from ref. 13, where the best-fit strain amplitude is at the edge of the values simulated from supermassive black hole binary population synthesis models. The inferred strain amplitude lies at the theoretical upper limit of the predicted astrophysical range[10,14,15]. It might also be in tension with the observed black hole mass function[16,17], although see[18] for a different view. Furthermore, the strain spectral index of the gravitational wave background is in about $2\sigma$ tension with the value corresponding to binary inspirals driven by gravitational wave emission alone ($\gamma = 13/3$, $\alpha = 2/3$)[19]. The tension is visible in Fig. 5 in ref. 13 and Fig. 11 in ref. 5, where the posterior on the spectral index is compared not with a single value, but with the distribution of values expected from realistic backgrounds made up of discrete point sources. Although previous PTA results were consistent with a very broad range of assumptions about binary black hole populations[20], they suggested deviations from purely gravitational wave-driven binary evolution. Furthermore, deviations of the strain spectral index from $-2/3$ can point to the early-Universe origin of the signal[13,21].

Modelling of pulsar-intrinsic noise is important because it can affect the conclusions about the properties of the gravitational wave background, such as $A$ and $\gamma$[22]. It is relevant for the European Pulsar Timing Array (EPTA), one of the world's leading PTAs, where the latest 10-year data set showed evidence for the gravitational wave background at $3.5\sigma$[6]. Despite dedicated noise modelling studies being performed for the EPTA[23], there are four indications that some noise is still mismodelled. First, it was pointed out in several studies[2,4,24] that the standard PTA models of how noise parameters are distributed across pulsars are incorrect. These models manifest as prior probabilities in PTA data analyses. To be precise, the models are incorrect because they are 'static', *i.e.*, the shape of the distribution of noise parameters is not influenced by the data. Although imposing such priors is very unlikely to influence our conclusions about evidence for the gravitational background[25–28], it may introduce systematic errors in the inferred strain spectrum[29]. Second, epoch-correlated temporally-uncorrelated (white) noise term[30] was neglected in the previous EPTA analysis. Third, van Haasteren[31] pointed out that the procedure of removing certain noise terms in the search for gravitational waves based on the results of single-pulsar noise analysis, performed in the original EPTA analysis[6], is prone to systematic errors. Fourth, the original EPTA analysis treats the transient noise effect in pulsar PSR J1713 + 0747 as from a sudden change in dispersion measure, although Goncharov et al.[32] suggested that it is associated with the pulse shape change.

In this work, we address the above four limitations and revise the properties of the gravitational wave background inferred by the European Pulsar Timing Array (EPTA)[6]. As a result, we find a steeper characteristic strain spectrum of the background, which is in better agreement with the hypothesis that the background originates from a superposition of adiabatically inspiraling supermassive binary black holes in circular orbits. Based on the revised amplitude and spectral index of the background, we discuss implications for the evolution of supermassive black hole binaries. Finally, we describe the observational impact of our new noise model. In particular, we show that despite our new model results in less evidence for Hellings-Downs correlations, evidence remains strong. Our results highlight the importance of accurate noise models for correctly inferring background properties.

## Results

Posterior distributions for $A$ and $\gamma$ are shown as contours in Fig. 1. Solid blue contours correspond to our improved model. For comparison, dashed red contours correspond to the noise model used in the original analysis of the EPTA data[6]. The results are shown for both the 10-year subset of the EPTA data, which showed evidence for the Hellings-Downs correlations, and the full 25-year EPTA data, where the evidence is not visible[33]. The value of $\gamma = 13/3$ ($\alpha = 2/3$) is shown as a dashed straight line. Our improved model results in a lower median-*aposteriori* strain amplitude of the gravitational wave background, as well as in a steeper spectral index, as shown with solid blue contours in

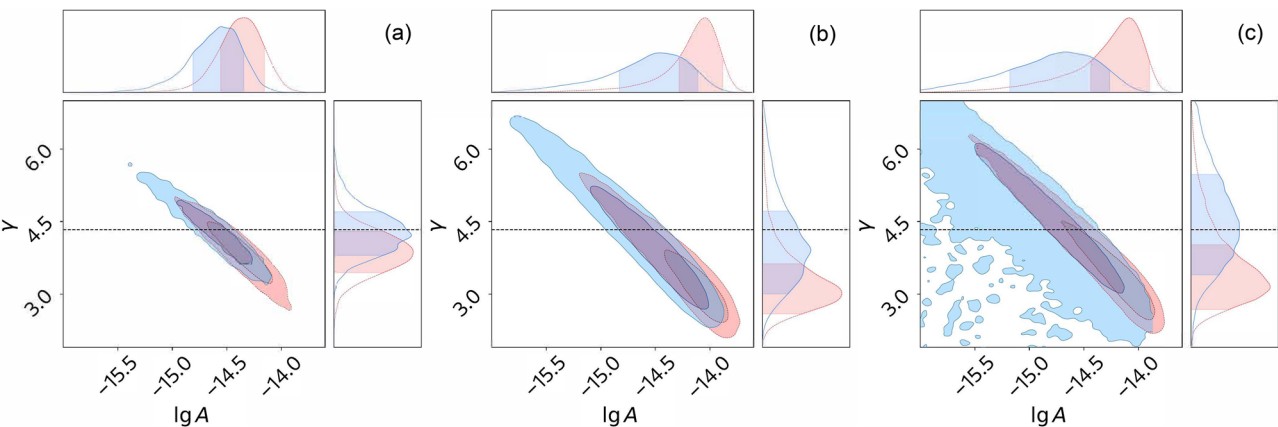

**Fig. 1 | Posterior distributions for the power-law amplitude $A$ and spectral index $\gamma$ of the putative gravitational wave background in the European Pulsar Timing Array (EPTA) data.** The results of a fit to Hellings-Downs correlations are shown in the left panel (**a**, full 25-year data, $(\lg A, \gamma) = (-14.53^{+0.15}_{-0.27}, 4.14^{+0.56}_{-0.26})$) and the middle panel (**b**, the 10-year subset of the data described in ref. 6, $(\lg A, \gamma) = (-14.37^{+0.17}_{-0.57}, 3.38^{+1.50}_{-0.25})$). The results of a fit of only temporal correlations to the 10-year data are shown in the right panel (**c**, $(\lg A, \gamma) = (-14.52^{+0.15}_{-0.80}, 4.20^{+1.32}_{-0.72})$). Dashed red contours correspond to the result using the standard pulsar noise priors, and the blue contours correspond to our improved model. The horizontal dashed line corresponds to the background from supermassive binary black holes inspiralling entirely due to gravitational wave emission (subject to cosmic variance). Our improved model results in a lower median-*aposteriori* strain amplitude of the background and mitigates tensions with $\gamma = 13/3$. In marginalised distributions, shaded areas correspond to $1\sigma$ credible levels. In joint distributions, an inner dark area corresponds to the $1\sigma$ level, and the outer lighter area corresponds to the $2\sigma$ level.

Fig. 1. Maximum-*aposteriori* (lg$A$, $\gamma$) obtained with our improved model and the respective measurement uncertainties based on 1$\sigma$ credible levels are reported in the caption of Fig. 1.

A detailed inspection of the contribution of four new components of our noise models, not shown in Fig. 1, reveals the following. Using noise model averaging instead of model selection, as recommended by ref. 31, has mostly affected the measurement uncertainty. Every other component of our improved noise model results in a consistently lower background strain amplitude and a steeper spectral index. Resolving noise prior misspecification using hierarchical inference[24], the first component of our improved noise model, has mostly affected the 25-year data (Fig. 1a) and the 10-year data when inter-pulsar correlations are not modelled (Fig. 1c). When modelling Hellings-Downs correlations in the 10-year data (Fig. 1b), other components of our revised noise model have made most of the impact. In particular, we found that the amplitude of the transient noise event in PSR J1713+0747 depends on the radio frequency consistently with ref. 32, not as assumed in the original EPTA analysis[6]. Modelling a transient noise event in PSR J1713 + 0747 correctly results in a significant shift in the posterior. Finally, epoch-correlated noise has not been considered in previous EPTA analyses because of a low number (1–4) of pulse arrival time measurements per observation epoch compared to other PTAs[6,23]. Nevertheless, we find evidence for this noise in the data from certain backend-received systems. In Fig. 1b, the choice leads to the consistency of the fully marginalised posterior on $\gamma$ with 13/3 at the 1$\sigma$ level. Revising noise priors alone, without considering epoch-correlated noise, yields a weaker consistency of the joint posterior on (lg$A$, $\gamma$) with $\gamma$ = 13/3 at the 1$\sigma$ level (not shown in Fig. 1b). Stronger impact of hierarchical inference in Fig. 1a, c compared to Fig. 1b illustrates how inter-pulsar correlations influence the posterior.

## Implications for supermassive binary black holes

If the energy loss in binary inspirals is dominated by the adiabatic emission of gravitational waves and if binaries are circular, the characteristic strain spectrum of the gravitational wave background is ref. 19

$$h_c^2(f) = \frac{4G^{5/3}}{3\pi^{1/3}c^2} f^{-4/3} \int \frac{d^2N}{dVdz} \frac{\mathcal{M}^{5/3}}{(1+z)^{1/3}} dz, \quad (1)$$

where $G$ is the Newton's constant, $c$ is the speed of light, $z$ is redshift, $\mathcal{M}$ is the binary chirp mass, and $d^2N/(dVdz)$, a function of $(\mathcal{M}, z)$, is the number density of binaries per unit comoving volume per unit redshift. The integral does not depend on a gravitational wave frequency, thus $h_c \propto f^{-2/3}$, as stated earlier. The background amplitude $A$ depends on the mass spectrum and the abundance of supermassive binary black holes in the universe. The derived value of $\alpha = 2/3$ ($\gamma = 13/3$) is confirmed by population synthesis simulations, *e.g.*, Figure 7 in ref. 12, where the theoretical uncertainty is only $\delta\gamma$ of approximately 0.1 at 1$\sigma$ due to cosmic variance[34,35].

Tensions of $\gamma$ of 3 inferred from the previous EPTA analysis with 13/3 = 4.3(3) reported during the announcement of evidence for the gravitational background[5–8] has led to discussions on whether the signal is influenced by certain effects of binary evolution that make the strain spectrum to appear flatter. Mechanisms of flattening $h_c(f)$ typically involve the introduction of a more rapid physical mechanism of a reduction in binary separation compared to a gravitational wave emission at < 0.1 parsec separations. Such a mechanism could be an environmental effect[36], such as stellar scattering[37,38], the torques of a circumbinary gas disc[39]. Furthermore, it could be due to the abundance of binaries in eccentric orbits, which leads to a more prominent gravitational wave emission[40]. Although eccentricity also results in a steeper $h_c(f > 10^{-8}\mathrm{Hz})$[41], but PTA sensitivity declines towards high frequencies. In contrast, the results of our improved analysis are consistent with binary evolution driven only by the emission of gravitational waves.

Our improved model also impacts the measurement of the strain amplitude, which is directly related to the number density of supermassive black hole binaries. The new results suggest that the supermassive black hole binaries are not as (over-)abundant as the earlier measurements implied. This is shown in Fig. 2, the bottom panel of which is based on Figure A1 in ref. 12. Grey horizontal bands correspond to theoretical uncertainties on the strain amplitude at the 16th - 84th percentile level in 26 studies[10,15,42–65]. For ref. 50 the model is "HS-nod", and for ref. 51 the model is "HS_nod_SN_high_accr". Our improved model reduces tensions of the gravitational wave background strain amplitude with theoretical and observationally-based predictions for supermassive black hole binaries. Furthermore, the revised background strain amplitude corresponds to longer delay times between galaxy mergers and supermassive binary black hole mergers[51]. The caveat is that the reported amplitude is referenced to $f$ = yr$^{-1}$, so the covariance between lg$A$ and $\gamma$ in posterior changes following a rotation of a power law about this frequency. We tested that at $f$ = 10yr$^{-1}$ our improved model mostly affects the posterior on $\gamma$, introducing consistency with 13/3, leaving the posterior on lg$A_{10yr}$ almost unchanged. Precisely, with 10-year EPTA data, we measure (lg$A_{10yr}$, $\gamma$) to be $(-14.02^{+0.08}_{-0.11}, 4.35^{+0.42}_{-1.02})$ with our new noise model and $(-14.00^{+0.08}_{-0.13}, 3.02^{+0.89}_{-0.25})$ with the original model. Consistently, one may notice in Figure A1 in ref. 12 that the posterior on lg$A_{10yr}$ referenced to $f$ = 10yr$^{-1}$ are in less tension with theoretical predictions.

As shown in ref. 16, Equation (1) can be presented as a function of black hole number density $\rho_{BH}$, black hole mass scale $M_*$, and the intrinsic scatter $\epsilon_0$ in galaxy stellar velocity dispersion, which is a proxy of mass for supermassive black holes,

$$h_c^2(f) = 1.2 \times 10^{-30} \left(\frac{f}{\mathrm{yr}^{-1}}\right)^{-4/3} \times \left(\frac{M_*}{5.8 \times 10^7 M_\odot}\right)^{2/3}$$
$$\times \left(\frac{\rho_{BH}}{4.5 \times 10^5 M_\odot \mathrm{Mpc}^{-3}}\right) \times \left(\frac{2}{e^{\frac{8}{9}\epsilon_0^2 \ln^2(10)}}\right). \quad (2)$$

The mass scale $M_*$ is expressed as $\lg M_* = a_\bullet + b_\bullet \frac{\sigma_*}{200\mathrm{kms}^{-1}}$. It is also worth noting that $\rho_{BH} \propto M_*$. Based on the equation above, we perform parameter estimation on ($\rho_{BH}$, $a_\bullet$, $b_\bullet$, $\sigma_*$) with the 10-year EPTA data and Hellings-Downs correlations. We assume Normal priors for ($a_\bullet$, $b_\bullet$, $\sigma_*$) as well as the value of $\epsilon_0$ = 0.38 from ref. 16. The priors for ($a_\bullet$, $b_\bullet$) for black hole masses are inferred from kinematic observations of local black holes in galaxy catalogues[66]. The prior for $\sigma_*$ is based on the velocity dispersion measured in the Sloan Digital Sky Survey[67]. We adopt a Normal prior on $\rho_{BH}$ from ref. 18 because a prior from ref. 16 yields the strain amplitude in significant tension with the EPTA observations. The posterior and the prior are shown in Fig. 3. All parameters are degenerate with each other due to representing a single observed quantity, the strain amplitude. Nevertheless, the figure illustrates which properties of the supermassive black hole population are constrained the most by the pulsar timing data, given astrophysical uncertainties. Provided the scaling in Equation (2), (0.6%, 5.7%, 0.9%) uncertainties on ($a_\bullet$, $b_\bullet$, $\sigma_*$), which contribute to $M_*$, and a larger 44.4% uncertainty on $\rho_{BH}$, the posterior informs solely on the number density $\rho_{BH}$. Our improved model influences the posterior to a lesser extent compared to Fig. 1 because the spectral index is fixed at $\gamma$ = 13/3. Accordingly, it can be noticed in Fig. 1 that lg$A$ does not change much for our improved noise model at the slices of fixed $\gamma$ = 13/3.

## Discussion

When the model closely matches reality, one would expect a reduction of the measurement uncertainty when adding extra data. This is not visible in the original EPTA analysis, where the 1$\sigma$ range for (lg$A$, $\gamma$) spans 0.44 for the 10-year data and 0.39 for the 25-year data before our

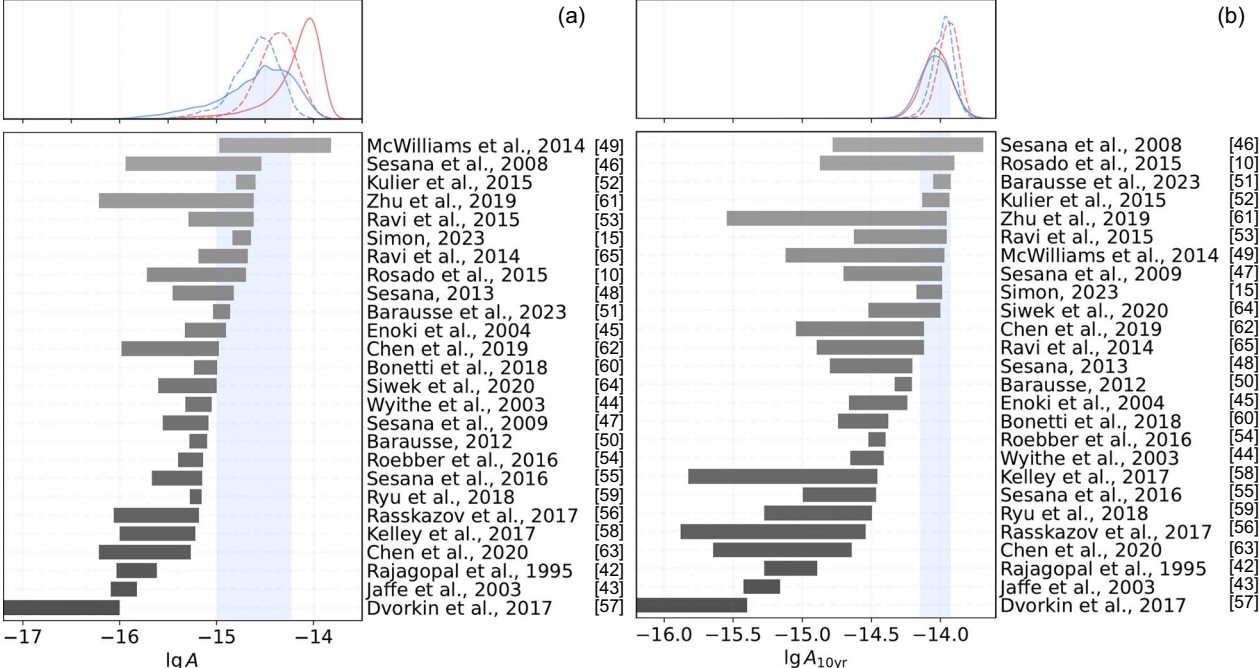

**Fig. 2 | Consistency of the inferred strain amplitude of the gravitational wave background with theoretical expectations.** Bottom panels show the predicted log-10 strain amplitude lg$A$ from 26 studies. The top panels show posteriors on lg$A$ marginalised over $\gamma$. Red lines correspond to the original noise model, and blue lines correspond to our revised model. Solid lines correspond to the 10-year data, and dashed lines correspond to the 25-year data. Blue bands correspond to 1$\sigma$ credible levels for 10-year data and our improved noise model. Panel (**a**) shows the characteristic strain amplitude at yr$^{-1}$, lg$A$. Panel (**b**) shows the strain amplitude at the frequency of (10yr)$^{-1}$.

improved noise model. As shown in Fig. 1c, our improved analysis yields 0.95 for the 10-year data and 0.42 for the 25-year data. A larger reduction in the measurement uncertainty when adding ten extra years of data is in agreement with our expectations. A decrease in 1$\sigma$ uncertainty levels in Fig. 1c compared to Fig. 1b indicates that inter-pulsar correlations in the 10-year data provide additional constraints on the background amplitude and spectral index. A larger maximum-*aposteriori* lg$A$ in Fig.1b compared to both Fig. 1a and Fig. 1c may also be driven by inter-pulsar correlations.

The shift of the posteriors in Fig. 1 towards larger spectral indices and smaller amplitudes with our improved analysis suggests that the louder pulsar-intrinsic noise with flatter spectra leaks into our measurement of the background strain spectrum when the four new sources of noise we point out are not modelled. The covariance between lg$A$ and $\gamma$ in Fig. 1 is along the line of equal noise power. Furthermore, in the early part of the 25-year data, our improved model may ameliorate instrumental noise and a lack of frequency coverage in addition to better modelling of millisecond pulsar spin noise[33].

Because the 10-year data and the 25-year data are not independent data sets, a high degree of consistency is expected. In the original EPTA analysis, maximum-*aposteriori* (lg$A$, $\gamma$) in the 25-year data differ from those of the 10-year data by (0.29, 0.76). It is visible in red contours across all three panels in Fig. 1. A smaller difference between the best-fit (lg$A$, $\gamma$) in the 25-year data and in the 10-year data of (0.16, 0.76) is achieved with our improved analysis. When not modelling Hellings-Downs correlations (Fig. 1c), the difference between the best-fit (lg$A$, $\gamma$) in the 25-year data and in the 10-year data is only (0.01, 0.06). Because the best-fit (lg$A$, $\gamma$) obtained with only temporal correlations is still expected to match those obtained with temporal and Hellings-Downs correlations, it is also possible that we have not removed all noise model misspecification from the analysis of the EPTA data.

Let us briefly hypothesise about the nature of any other mis-modeled noise. We note that the North American Nanohertz Observatory for Gravitational Waves (NANOGrav) has mitigated a tension of

the background spectral index with 13/3 by adopting the Gaussian process model of the dispersion variation noise[5], as in the EPTA analysis[23]. Therefore, one potential source of a systematic error could be the mismodelling of the pulsar-specific noise that depends on the radio frequency. A very nearby binary is another example[68–71]. Frequency-wise comparison of the inferred strain spectrum against black hole population synthesis models performed earlier by the EPTA (Fig. 3 in ref. 13) suggests that the deviation from $\gamma = 13/3$ may occur due to excess noise in two frequency bins, $1.3 \times 10^{-8}$ Hz and $2.9 \times 10^{-8}$ Hz. The rest of the spectrum appears to be consistent with $\gamma = 13/3$. The aforementioned potential sources of systematic errors may require better temporal and inter-pulsar correlation models of the data as part of future work.

Finally, we calculate evidence for Hellings-Downs correlations in the EPTA data with the revised noise model. Namely, the Bayes factor $\mathcal{B}$ in favour of the hypothesis that all pulsars contain Hellings-Downs correlated signal with ($A$, $\gamma$) against the hypothesis of the same signal without Hellings-Downs correlations. For 10-year EPTA data, we find $\mathcal{B} = 38$ (ln $\mathcal{B} = 3.63$). For the original EPTA noise model, we find a higher value of $\mathcal{B} = 59$, consistent with the previously reported value of 60 in Table 5 in ref. 6. Therefore, our improved model has slightly decreased evidence for Hellings-Downs correlations. It is a combination of the increased measurement uncertainty and, potentially, a fraction of the previously-reported evidence due to the propagation of unmodelled noise into evidence. For 25-year EPTA data, we find $\mathcal{B} = 3$ (ln $\mathcal{B} = 1.17$). For the original EPTA noise model, we find a consistent value ($\mathcal{B} = 3$, ln $\mathcal{B} = 0.93$), which is similar to the previously reported value $\mathcal{B} = 4$ from Table 5 in ref. 6.

To evaluate the broad form of inter-pulsar correlations in the 10-year EPTA data, we present Fig. 4. It shows the posterior on inter-pulsar correlations, as a function of angular separation between EPTA pulsar pairs, with our revised noise model. The result shows broad consistency with Hellings-Downs correlations. With this, we should point out a caveat that some of the parameter space in correlation

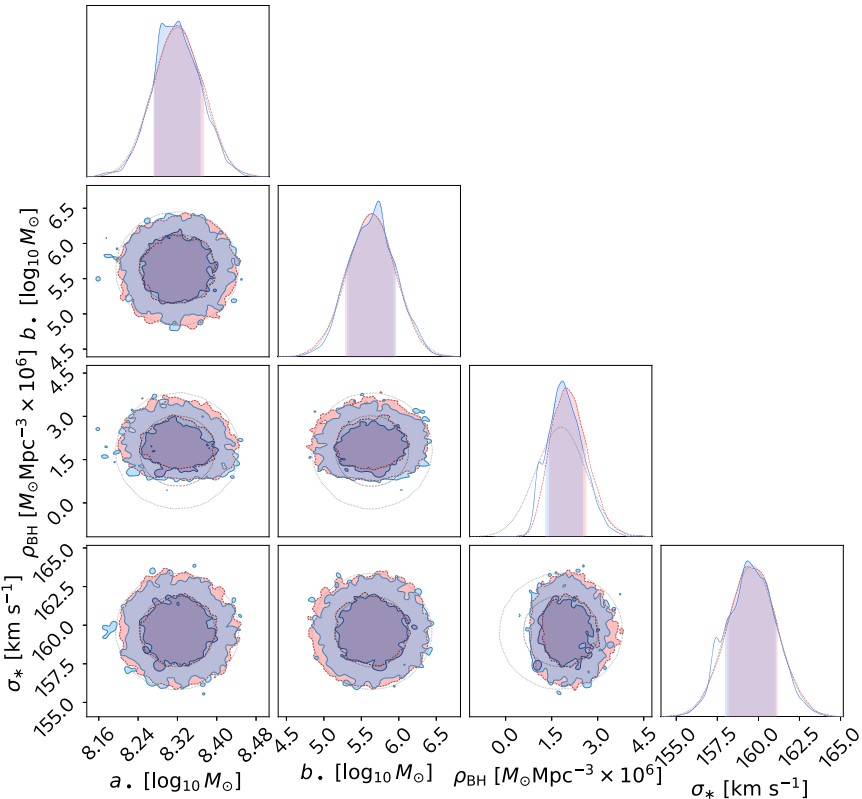

**Fig. 3 | Posterior of the population parameters for supermassive black hole binaries with the 10-year data.** Effectively, the constraints are provided solely by the inferred characteristic strain amplitude of the gravitational wave background, assuming the strain power law index of −2/3. Hellings-Downs correlations are included in the model.

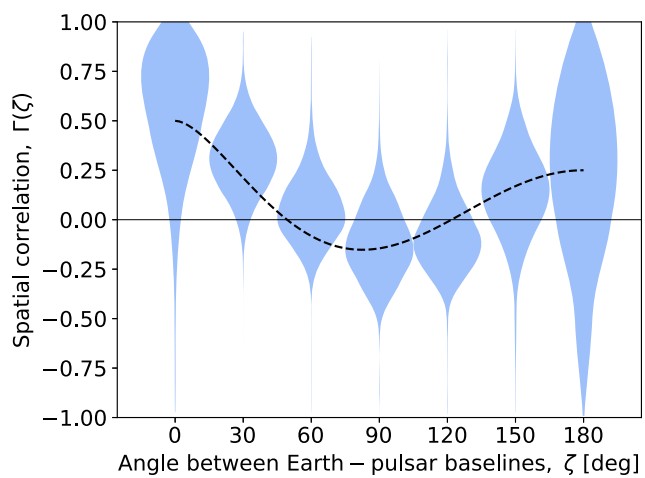

**Fig. 4 | Posterior on inter-pulsar correlations of the common stochastic process in the 10-year data.** Here, we assume our improved noise model. Correlations are shown as a function of angular separation between the observed pulsar pairs. Hellings-downs function of the gravitational wave background is shown as the dashed line.

coefficients may be ruled out solely because they do not result in a positive definite covariance matrix of the likelihood. Thus, strictly speaking, flexible correlation models are ill-posed (Rutger van Haasteren, private communication).

## Methods

PTAs perform precision measurements of pulse arrival times from millisecond radio pulsars. The likelihood of delays(-advances) $\delta t$ for a vector of pulse arrival times $t$ is modelled as a multivariate Gaussian distribution $\mathcal{L}(\delta t|\theta)$, where $\theta$ is a vector of parameters of models that describe the data. From Bayes' theorem, it follows that the posterior distribution of model parameters is $\mathcal{P}(\theta|\delta t) = \mathcal{Z}^{-1}\mathcal{L}(\delta t|\theta)\pi(\theta)$, where $\mathcal{Z}$ is Bayesian evidence, a fully-marginalised likelihood. The term $\pi(\theta)$ is called a prior probability distribution, a model of how likely it is to find a certain value of $\theta$ in Nature. Model selection is performed by computing the ratio of $\mathcal{Z}$ for pairs of models, which is referred to as the Bayes factor. The Bayes factor is equal to the Bayesian odds ratio if both models are assumed to have equal prior odds.

The time-domain likelihood of the observed radio pulsar pulse delays-advances time series $\delta t$, at the position and of the Solar System barycenter, is a multivariate Gaussian,

$$\mathcal{L}(\delta t|\theta) = \frac{\exp\left(-\frac{1}{2}(\delta t - \mu)^{\mathsf{T}} C^{-1}(\delta t - \mu)\right)}{\sqrt{\det(2\pi C)}}. \tag{3}$$

Here, $\theta$ are model parameters. Vector $\mu$, a function of $\theta$, represents the model prediction for timing residuals. The covariance matrix is $C = N + T^{\mathsf{T}}B^{-1}T$. The likelihood is marginalised over coefficients $b$ which determine the time series realisation $Tb$ of the corresponding signal or noise. Square matrix $N$ only contains diagonal elements that represent temporally uncorrelated noise, which is referred to as white. The so-called design matrix $T = [M, F, U]$ is made up of three blocks: for the timing model, $M$, for the Fourier series of time-correlated signals, $F$, and for the epoch-correlated noise, $U$. Design matrix maps the model parameter space (columns) to the pulse time of arrival space (rows), such that $Tb$ is the time series, like $\mu$. Precisely, parameters $\epsilon$ for $M$ are those of the EPTA timing model[72], including pulsar spin frequency, its derivative, pulsar sky position, *etc*. Parameters for $F$ are power spectral

density amplitudes $a$ corresponding to Fourier sine and cosine terms at Fourier frequencies. We model temporal correlations in 30 frequencies for a common time-correlated stochastic signal attributed to the gravitational wave background; the number of frequencies for time-correlated noise is determined based on single-pulsar noise analysis[23]. Parameters for $U$ are delay time series for every epoch, $j$; epochs are groups of nearly simultaneous pulse arrival times at different radio frequencies. Coefficients $b = [\epsilon, a, j]$, corresponding to the same signals as in $T$, represent signal amplitudes before a transformation to the time domain. Finally, the covariance matrix of coefficients, $B$, is such that the coefficients $b$ can be generated from the zero-mean Gaussian distribution described by the covariance matrix $B$. It is the matrix with diagonal blocks $[\xi, \varphi, \mathcal{J}]$. The first component $\xi$, a diagonal matrix of the values of $10^{40}$, corresponds to a wide, uninformative prior on the timing model coefficients $\epsilon$. Component $\mathcal{J}$ is the covariance matrix of white epoch-correlated noise. The second component, $\varphi$, is such that

$$\varphi_{(ai),(bj)} = P_{ai}\delta_{ab}\delta_{ij} + \Gamma_{ab}P_i\delta_{ij}, \qquad (4)$$

where $(a, b)$ are pulsar indices, $(i, j)$ are frequency indices, $P_{ai}$ is the power spectral density of noise, $P_i$ is the power spectral density of a common time-correlated signal, and $\Gamma_{ab}$ determines the degree of spatial correlations between pulsars $a$ and $b$. In the case of an isotropic gravitational wave background from black hole binaries in circular orbits, $\Gamma_{ab}$ is given by the Hellings-Downs function[9]:

$$\Gamma_{ab}|_{a\neq b} = \frac{1}{2} - \frac{x_{ab}}{4} + \frac{3}{2}x_{ab}\ln x_{ab}, \qquad (5)$$

where $\zeta_{ab}$ is the sky separation angle for a given pair of pulsars and $x_{ab} = (1 - \cos\zeta_{ab})/2$. In addition, $\Gamma_{aa} = 1$.

Time-correlated noise, which yields stochastic timing delays, can be categorised into achromatic and chromatic. Achromatic noise does not depend on the pulsar's pulse radio frequency. It is also referred to as spin noise because it represents pulsar rotational irregularities. Chromatic noise[73] is characterised by an amplitude that depends on the radio frequency at which it is observed. Most EPTA pulsars have measurable levels of dispersion measure (DM) noise, and many pulsars are shown to have spin noise[23]. Epoch-correlated noise, associated with pulse jitter (short-term pulse profile variations), can influence the inferred strain spectrum of the gravitational wave background at the highest frequencies because of the "white" nature of this noise. Sudden dips with exponential relaxations in timing delays, associated with pulse shape changes on timescales of days-months[32], can also influence parameter estimation for time-correlated signals if not modelled. Noise terms missed in the model, despite evidence, are expected, in many cases, to incorrectly increase the inferred signal amplitude. In particular, the gravitational wave model would attempt to absorb excess noise power.

The priors are our models of how parameters governing pulsar-specific noise are distributed across the pulsars. PTA data provides information about the distribution of $\theta$ in Nature, so failing to model this distribution could lead to prior misspecification. To address this, we parametrise prior distributions for amplitudes and spectral indices of pulsar spin and DM noise spectra. Our improved analysis introduces hyperparameters $\Lambda$ to parametrise priors: $\pi(\theta|\Lambda)\pi(\Lambda)$. We then perform a numerical marginalisation over $\Lambda$. We use a new procedure of marginalisation over hyperparameters, which is described in Section 2.2.2 of the companion paper[24].

Because the gravitational wave background is a stochastic time-correlated signal, it is important to hierarchically model other stochastic time-correlated signals in the data. Pulsar spin noise is described by amplitudes and spectral indices ($A_{\mathrm{SN}}$, $\gamma_{\mathrm{SN}}$), which are different for every pulsar. Similarly, DM noise is characterised by ($A_{\mathrm{DM}}$, $\gamma_{\mathrm{DM}}$),

where the amplitude is referenced to 1400 MHz. Hierarchical noise model directly impacts posteriors on ($A_{\mathrm{SN}}$, $\gamma_{\mathrm{SN}}$, $A_{\mathrm{DM}}$, $\gamma_{\mathrm{DM}}$). Because the data is only compatible with a certain total amount of timing fluctuations, hierarchical noise inference also impacts posteriors on the gravitational wave background's amplitude.

## Data availability
Posterior samples generated in this study have been deposited in the database at zenodo.org/record/15716517[75]. Second data release of the European Pulsar Timing Array[72] is available at zenodo.org/record/8091568[76] and gitlab.in2p3.fr/epta/epta-dr2.

## Code availability
The code to marginalise over the uncertainties in pulsar noise priors is available at github.com/bvgoncharov/pta_priors (commit "96cd7cc" is used for the analysis presented in this work). The PTA likelihood is incorporated in ENTERPRISE[77], and posterior sampling is performed using PTMCMCSAMPLER[78].

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

## Acknowledgements
We thank Bruce Allen for contributions to the structuring of the early version of this manuscript. We thank Rutger van Haasteren for discussions about hierarchical Bayesian inference. We thank Emily Liepold and other colleagues for helpful comments on the manuscript. We thank Gabriela Sato-Polito for clarifying the details of ref. 16. This research was supported in part by a grant NSF PHY-2309135 to the Kavli Institute for Theoretical Physics (KITP). Some of our calculations were carried out using the OzSTAR Australian national facility (high-performance computing) at Swinburne University of Technology. European Pulsar Timing Array is a member of the International Pulsar Timing Array[74]. J.A. acknowledges support from the European Commission (ARGOS-CDS; Grant Agreement number: 101094354). A.C. acknowledges financial support provided under the European Union's Horizon Europe ERC Starting Grant "A Gamma-ray Infrastructure to Advance Gravitational Wave Astrophysics" (GIGA; Grant Agreement: 101116134).

## Author contributions
B.G. performed the analysis and paper writing. S.S. found the optimal hierarchical model for time-correlated noise and contributed to the figures. A.S. made suggestions on the astrophysical interpretation of the results. S.M.T. found evidence for epoch-correlated noise in the data, which is included in our revised noise model. Other authors listed alphabetically – J.A., A.C., D.J.C., S.C., E.F.K., K.L., G.S., L.S., and S.V. – have performed pulsar observations or otherwise contributed to the development of the data release used in this work.

## Funding

## Competing interests
The authors declare no competing interests.
