## [Transparent Peer Review file · Nature Communications]

Reading signatures of supermassive binary black holes in pulsar timing array observations

Corresponding Author: Dr Boris Goncharov

Version 1:

Reviewer comments:

Reviewer #1

(Remarks to the Author)

This work provides an important revision of prior estimates of the GWB from the EPTA data. Overall, this is a nice paper which is well written, concise, and certainly warranting of publication.

I do have a few somewhat significant concerns to address:

1. It doesn't seem that the title is representative of the main work done here and the main conclusion presented in the title (**fewer** supermassive binary black holes...) is not actually directly justified in the text. Section II discusses improved posteriors in A and γ from the 25yr and 10yr datasets with the improved noise models, overall resulting in a lower inferred strain compared to the prior model. That amplitude is certainly related to the number density of binaries (as in your eqn 1), but also to the mass scale and the redshift where those binaries emit their GW waves. As discussed in Sato-Polito+23 and Liepold+24, the strong dependence on the mass scale (and underlying uncertainties in that scale in the high-mass regime which contribute to the bulk of the GW background) suggest substantial uncertainties in the number density given a fixed amplitude. A reduction in the mass scale or number density of binaries would be consistent with the reduced amplitude found here, but the title and discussion in II.A seem to take the mass scale of these binaries to be well established when in fact there is substantial discussion of the topic in the literature (also see Bernardi+13, D'Souza+15, Leja+20) for other considerations of the major uncertainties in galaxy stellar masses and dispersions which are used to infer binary BH masses. If the title of this manuscript is to stay as it is, it would be vital to quantify or estimate how many fewer SMBHB are expected given the new reduction of the EPTA data.

2. While the format of Nature Communications values brief manuscripts which rapidly disseminate results, the provided manuscript seems perhaps too terse. The primary result here which does not appear in the companion paper (Goncharov+24) is the application of the improved noise models discussed there to yield posteriors for the amplitude and spectral index. The connection between the noise model (and the nuances of that model) and the inferred physical quantities is non-trivial and potentially a very consequential result which could be gleaned from this work. Section II more or less notes that changing the noise model changes the posterior but neither that section nor section III appear to discuss the physical (or computational, etc) connection between those two disparate stages of the analysis. This work is well suited to examine or discuss that connection. Some of this connection could be elucidated through the methods section which at the moment just has a broad discussion of Bayesian statistics. A brief discussion of how the amplitude / spectral index may be related to the PTA data and depends on the noise model would be extremely valuable. Of course details can still be left to the companion paper but at the moment the methods section doesn't appear to provide new information beyond the reference to the companion.

3. Similar to concern 2, this manuscript seems to lack sufficient discussion of the context of its results. For example, in the discussion section it is noted that the uncertainties are reduced between the 10yr and 25yr datasets and it is noted that when changing from the prior EPTA analysis to the current one reduces the difference between the 10yr and 25yr posteriors. But it doesn't seem that the context of those uncertainties or discrepancies is discussed, nor are consequences of those differences. For example, does the improved agreement between the 10yr and 25yr posteriors change predictions of other physical features? Does the improved amplitude and index posterior change the agreement / disagreement with different models or analyses of the origin of the GWB? Section II.A appeared to start this sort of inquiry, but a more complete discussion of this and other context is essential.

Emily Liepold

(Remarks on code availability)

I have reviewed the code, and it appears to be well-structured and relatively easily installed. I was able to download and run the scripts (though I did not run any full analyses using real data). The code appears to be relatively well documented, including a README file and examples.

Reviewer #2

(Remarks to the Author)

Overall Comments.

Methodologically, the paper represents an intellectual advance by applying a novel, physically-motivated, and data-appropriate model for ensemble pulsar noise properties (e.g. ref [19]) to the most recent EPTA 10-year and 25-year dataset. Applying this new model results in a slightly lower gravitational wave background amplitude using the 25-year dataset, with the implication that the previous higher value resulted from excess noise leaking into the measured gravitational wave background signal. The higher Bayes factor for the Hellings-and-Downs cross-correlated model using the improved noise model verifies that the new measurement is more likely to result from an isotropic gravitational wave background origin than the old measurement. Although the reduction in the GWB amplitude is small, small changes to the observed GWB amplitude imply large changes to the number density of supermassive black holes since $A \propto \sqrt{N}$. As such, the updated measurements will be useful for the community. This result also reduces the apparent tension between observed and predicted SMBHB populations. As such, the paper is suitable for publication Nature Communications.

Major comments:

- It seems as if the ensemble model improves the measurement of SMBHBs in the EPTA 25-year dataset, more so than the 10-year dataset. The authors should provide some explanation as to why the 10-year measurements are not improved when using the ensemble noise model and HD-correlated GWB model, as shown in Figure 1b.

- It should also be discussed that if there is excess noise from the older backend-receiver systems of the telescopes, as the authors indicate is a possibility, then this result suggests that the ensemble noise model is potentially ameliorating the common instrumental systematics of the early data more so than the intrinsic spin noise properties of MSPs.

- Observational impact: To allow readers to better understand the changes in significance of HD correlations, the authors should report the values of the Bayes factors and SNR distributions they obtained using each model and dataset in a table, similar to the original EPTA analysis.

Minor comments:

- In the introduction, when the authors state that “the strain spectral index of the gravitational wave background is in $\approx 2\sigma$ tension with the value corresponding to binary inspirals driven by gravitational wave emission alone,” they should probably specify that the value in question is an expectation value for the spectral index across many realizations of SMBHBs (which also neglects effects of discreteness due to individual binaries). It is not necessarily unexpected that the realization of the GWB in our universe might deviate from the expectation, e.g. based on the “very nearby binary” hypothesis the authors reference later on.

- Fig 1: In the 2D distributions, does the 1-sigma level enclose 68% of the probability, as for a 1D Gaussian, or 39% of the probability, as for a 2D Gaussian

(Remarks on code availability)

Reviewer #3

(Remarks to the Author)

This paper investigates the impact of improved noise models on the estimated parameters of the nanohertz gravitational wave background. The authors introduce four key enhancements, including addressing prior noise misspecifications and incorporating epoch-correlated noise leading to more accurate parameter estimation of the gravitational wave signal. This, importantly, can influence our understanding of the population of supermassive black hole binaries and their characteristics. The paper is well-written, clear, and impactful in the field, especially given the potential proximity to a 5-sigma detection of the background. As more data is added to existing datasets and more datasets from additional collaborations become available, these improved noise models and the implications of the EPTA findings will be crucial.

I only have minor comments:

1. While the paper clearly focuses on the background being produced by a population of supermassive black hole binaries, I believe it is important to acknowledge in the introduction that the background could also originate from cosmological sources. I believe this perspective is significant, especially considering that a paper like this (<https://arxiv.org/abs/2306.16219>) on the topic has been cited nearly 700 times.

2. A recent paper (<https://arxiv.org/abs/2502.04653>) shows that similar effects on the posterior estimates of the background, such as high amplitude and shallow spectral index, can arise from mismodelling of pulsar noise, particularly by omitting certain noise processes in the noise models. In this paper, the model is also enhanced by incorporating epoch-correlated white noise, which was not included in the EPTA previous analysis. I believe referencing this paper in the introduction would provide valuable context.

3. At the end of the "Results" section, the authors mention that they have tested a different reference frequency ($f = 10 \text{ yr}^{-1}$). This is an interesting point, and it would be very insightful to repeat Fig 2 with this reference frequency.

4. For clarity, in the "Method" section, second paragraph I would rephrase the sentence in the Method section:

"Because the total PTA noise prior is a product of noise priors for every pulsar, PTA data will inform on the distribution of θ in Nature. Neglecting this may result in prior misspecification."

to

"Since PTA data provides information about the distribution of θ in Nature, failing to account for this could lead to prior misspecification."

(Remarks on code availability)

Reviewer #4

(Remarks to the Author)

(Remarks on code availability)

Version 2:

Reviewer comments:

Reviewer #1

(Remarks to the Author)

With the authors' revisions, the concerns I expressed in my first report are now satisfied and I believe the manuscript is suitable for publication.

(Remarks on code availability)

Reviewer #2

(Remarks to the Author)

The authors have made a sincere attempt to address all my comments and suggestions. I believe this work is ready to be published.

(Remarks on code availability)

Reviewer #3

(Remarks to the Author)

Thank you for your detailed response and the revisions made to the manuscript. You have addressed all of my comments clearly and effectively.

In my view, the paper is now suitable for publication.

(Remarks on code availability)

Reviewer #4

(Remarks to the Author)

I co-reviewed this manuscript with one of the reviewers who provided the listed reports. This is part of the Nature Communications initiative to facilitate training in peer review and to provide appropriate recognition for Early Career

Researchers who co-review manuscripts.

(Remarks on code availability)

We thank the reviewers for their positive feedback. We took some time to respond because we have extended our hierarchical noise analysis to also include dispersion measure variation noise in addition to pulsar spin noise. Thus, we now accurately model all major sources of time-correlated (red) noise. The results have further supported our conclusions. With this, we believe we have addressed all comments in the revised manuscript. Please find more details below. Reviewer's points are highlighted by indentation.

=====

Response to Reviewer #1

This work provides an important revision of prior estimates of the GWB from the EPTA data. Overall, this is a nice paper which is well written, concise, and certainly warranting of publication

We thank Reviewer #1, Dr. Emily Liepold, for helpful comments and tests of our code. The report of Reviewer #1 contains three concerns. We explain how we have addressed them below.

Concern 1. Title, relating SMBHB number density and mass scale to the result.

1. It doesn't seem that the title is representative of the main work done here and the main conclusion presented in the title (****fewer**** supermassive binary black holes...) is not actually directly justified in the text. Section II discusses improved posteriors in A and γ from the 25yr and 10yr datasets with the improved noise models, overall resulting in a lower inferred strain compared to the prior model. That amplitude is certainly related to the number density of binaries (as in your eqn 1), but also to the mass scale and the redshift where those binaries emit their GW waves. As discussed in Sato-Polito+23 and Liepold+24, the strong dependence on the mass scale (and underlying uncertainties in that scale in the high-mass regime which contribute to the bulk of the GW background) suggest substantial uncertainties in the number density given a fixed amplitude. A reduction in the mass scale or number density of binaries would be consistent with the reduced amplitude found here, but the title and discussion in II.A seem to take the mass scale of these binaries to be well established when in fact there is substantial discussion of the topic in the literature (also see Bernardi+13, D'Souza+15, Leja+20) for other considerations of the major uncertainties in galaxy stellar masses and dispersions which are used to infer binary BH masses. If the title of this manuscript is to stay as it is, it would be vital to quantify or estimate how many fewer SMBHB are expected given the new reduction of the EPTA data.

We agree that the impact of our results on the number of SMBHBs is not obvious and that the title can be improved for clarity. Thus, we made the title broader in scope, replacing "Fewer" with "Reading signatures of". We have also acknowledged the impact of the mass scale on the interpretation in the abstract. Furthermore, we have added a new fourth

paragraph and Figure 3 to Section IIA with the discussion of the impact of black hole mass scale and number density on the observed strain amplitude.

Concern 2. The connection between the noise model and the inferred physical quantities, the methods section.

2. While the format of Nature Communications values brief manuscripts which rapidly disseminate results, the provided manuscript seems perhaps too terse. The primary result here which does not appear in the companion paper (Goncharov+24) is the application of the improved noise models discussed there to yield posteriors for the amplitude and spectral index. The connection between the noise model (and the nuances of that model) and the inferred physical quantities is non-trivial and potentially a very consequential result which could be gleaned from this work. Section II more or less notes that changing the noise model changes the posterior but neither that section nor section III appear to discuss the physical (or computational, etc) connection between those two disparate stages of the analysis. This work is well suited to examine or discuss that connection. Some of this connection could be elucidated through the methods section which at the moment just has a broad discussion of Bayesian statistics. A brief discussion of how the amplitude / spectral index may be related to the PTA data and depends on the noise model would be extremely valuable. Of course details can still be left to the companion paper but at the moment the methods section doesn't appear to provide new information beyond the reference to the companion.

We have significantly expanded the Methods section elucidating the connection between the noise models and the inferred physical quantities, as requested. In the abstract, we clarified the specific impact of our improved noise models on the strain amplitude and spectral index.

Concern 3. A lack of discussion of better agreement between 10-year and 25-year data, and the impact on other GWB analyses.

3. Similar to concern 2, this manuscript seems to lack sufficient discussion of the context of its results. For example, in the discussion section it is noted that the uncertainties are reduced between the 10yr and 25yr datasets and it is noted that when changing from the prior EPTA analysis to the current one reduces the difference between the 10yr and 25yr posteriors. But it doesn't seem that the context of those uncertainties or discrepancies is discussed, nor are consequences of those differences. For example, does the improved agreement between the 10yr and 25yr posteriors change predictions of other physical features? Does the improved amplitude and index posterior change the agreement / disagreement with different models or analyses of the origin of the GWB? Section II.A appeared to start this sort of inquiry, but a more complete discussion of this and other context is essential.

To elaborate on observational consequences of the improved noise modelling, we added a new fifth paragraph to Section III of the manuscript, which discusses the changes in evidence for the GWB. We have also mentioned the potential impact of our new noise models on the instrumental noise and a lack of radio frequency coverage in the "legacy" part of the 25-year data, in addition to the improved modelling of the intrinsic pulsar spin noise,

citing the relevant recent paper by Ferranti, Falxa, et al. (EPTA, 2025). Concerning physical quantities, we hope the new fourth paragraph in Section IIA and Figure 3 now provide a sufficient perspective.

Reviewer #1 (Remarks on code availability):

I have reviewed the code, and it appears to be well-structured and relatively easily installed. I was able to download and run the scripts (though I did not run any full analyses using real data). The code appears to be relatively well documented, including a README file and examples.

We thank the reviewer for testing our code.

=====

Response to Reviewer #2

Reviewer #2 (Remarks to the Author):

Overall Comments.

Methodologically, the paper represents an intellectual advance by applying a novel, physically-motivated, and data-appropriate model for ensemble pulsar noise properties (e.g. ref [19]) to the most recent EPTA 10-year and 25-year dataset. Applying this new model results in a slightly lower gravitational wave background amplitude using the 25-year dataset, with the implication that the previous higher value resulted from excess noise leaking into the measured gravitational wave background signal. The higher Bayes factor for the Hellings-and-Downs cross-correlated model using the improved noise model verifies that the new measurement is more likely to result from an isotropic gravitational wave background origin than the old measurement. Although the reduction in the GWB amplitude is small, small changes to the observed GWB amplitude imply large changes to the number density of supermassive black holes since $A \propto \sqrt{N}$. As such, the updated measurements will be useful for the community. This result also reduces the apparent tension between observed and predicted SMBHB populations. As such, the paper is suitable for publication Nature Communications.

We thank the reviewer for their valuable comments. We echoed some of the reviewer's summary points in the revised abstract. Reviewer #2 had three major concerns and two minor concerns.

Major concern 1. Explanation for Figure 1b.

Major comments:

- It seems as if the ensemble model improves the measurement of SMBHBs in the EPTA 25-year dataset, more so than the 10-year dataset. The authors should provide some explanation as to why the 10-year measurements are not improved when using the ensemble noise model and HD-correlated GWB model, as shown in Figure 1b.

To address this concern, we first added a new sentence at the end of the second paragraph of Section II: “Stronger impact ... illustrates how inter-pulsar correlations influence the posterior”. Next, we added a new sentence to the end of the first paragraph of Section III: “A larger maximum-a-posteriori $I_g A$ in Figure 1b compared to Figure 1a and Figure 1c may also be driven by inter-pulsar correlations”. An existing sentence at the end of the second paragraph of Section III further elaborates on this.

Major concern 2. Common instrumental systematics of the early data.

- It should also be discussed that if there is excess noise from the older backend-receiver systems of the telescopes, as the authors indicate is a possibility, then this result suggests that the ensemble noise model is potentially ameliorating the common instrumental systematics of the early data more so than the intrinsic spin noise properties of MSPs.

To address this concern, we added a new sentence to the end of the second paragraph of Section 3: “Furthermore, in the early part of the 25-year data, our improved model may ameliorate instrumental noise in addition to better modelling of millisecond pulsar spin noise.”

Major concern 3. Evidence for the GWB.

- Observational impact: To allow readers to better understand the changes in significance of HD correlations, the authors should report the values of the Bayes factors and SNR distributions they obtained using each model and dataset in a table, similar to the original EPTA analysis.

We add a new paragraph in Section III, where we report and discuss log Bayes factors which we obtain with the revised noise model. We have not calculated SNRs because they are redundant with Bayes factors, and we would prefer the manuscript to be more focused on SMBHBs.

Minor concern 1. Spectral index of 13/3 from SMBHBs and its realizations.

Minor comments:

- In the introduction, when the authors state that “the strain spectral index of the gravitational wave background is in $\approx 2\sigma$ tension with the value corresponding to binary inspirals driven by gravitational wave emission alone,” they should probably specify that the value in question is an expectation value for the spectral index across many realizations of SMBHBs (which also neglects effects of discreteness due to individual binaries). It is not necessarily unexpected that the realization of the GWB in our universe might deviate from the expectation, e.g. based on the “very nearby binary” hypothesis the authors reference later on.

We thank Reviewer #2 for pointing it out, we clarified it in the next sentence with references to EPTA and NANOGrav figures.

Minor concern 2. 39-60% probability levels.

- Fig 1: In the 2D distributions, does the 1-sigma level enclose 68% of the probability, as for a 1D Gaussian, or 39% of the probability, as for a 2D Gaussian

We thank Reviewer #2 for checking, we use 1-sigma levels consistent with the dimensionality throughout the manuscript.

=====

Response to Reviewer #3

Reviewer #3 (Remarks to the Author):

This paper investigates the impact of improved noise models on the estimated parameters of the nanohertz gravitational wave background. The authors introduce four key enhancements, including addressing prior noise misspecifications and incorporating epoch-correlated noise leading to more accurate parameter estimation of the gravitational wave signal. This, importantly, can influence our understanding of the population of supermassive black hole binaries and their characteristics. The paper is well-written, clear, and impactful in the field, especially given the potential proximity to a 5-sigma detection of the background. As more data is added to existing datasets and more datasets from additional collaborations become available, these improved noise models and the implications of the EPTA findings will be crucial.

We thank Reviewer #3 for helpful comments. The reviewer made four minor comments. We explain how we have addressed them below.

Comment 1. New physics interpretation of the GWB.

I only have minor comments:

1. While the paper clearly focuses on the background being produced by a population of supermassive black hole binaries, I believe it is important to acknowledge in the introduction that the background could also originate from cosmological sources. I believe this perspective is significant, especially considering that a paper like this (<https://arxiv.org/abs/2306.16219>) on the topic has been cited nearly 700 times.

We agree, we have now acknowledged this interpretation and cited the NANOGrav new physics paper in the third paragraph of the introduction.

Comment 2. Ref. 2502.04653.

2. A recent paper (<https://arxiv.org/abs/2502.04653>) shows that similar effects on the posterior estimates of the background, such as high amplitude and shallow spectral index, can arise from mismodelling of pulsar noise, particularly by omitting certain noise processes in the noise models. In this paper, the model is also enhanced by incorporating epoch-correlated white noise, which was not included in the EPTA previous analysis. I believe referencing this paper in the introduction would provide valuable context.

As suggested, we added a reference to the first sentence of paragraph 4 of the introduction, which seems most relevant.

Comment 3. Fig 2 with $f = 10 \text{ yr}^{-1}$.

3. At the end of the “Results” section, the authors mention that they have tested a different reference frequency ($f = 10 \text{ yr}^{-1}$). This is an interesting point, and it would be very insightful to repeat Fig 2 with this reference frequency.

We thank Reviewer #3 for this suggestion, we have implemented it. It now reveals interesting aspects of theoretical predictions. We have also sorted predictions by the highest value, to better highlight levels of their consistency with our results. We have also fixed a minor inaccuracy in the reported amplitude value at $f = 10 \text{ yr}^{-1}$ in the main text of the manuscript, it was due to a missing constant factor.

Comment 4. Rephrasing the second paragraph in the method section: “Since PTA data provides information about the distribution of θ in Nature, failing to account for this could lead to prior misspecification”.

4. For clarity, in the “Method” section, second paragraph I would rephrase the sentence in the Method section:

“Because the total PTA noise prior is a product of noise priors for every pulsar, PTA data will inform on the distribution of θ in Nature. Neglecting this may result in prior misspecification.”

to

“Since PTA data provides information about the distribution of θ in Nature, failing to account for this could lead to prior misspecification.”

We rephrased the sentence.

=====

Response to Reviewer #4

We thank Reviewer #4 for their participation.

=====